# Comparison of Pulse Wave Signal Monitoring Techniques with Different Fiber-Optic Interferometric Sensing Elements

**Nikolai Ushakov \***, **Aleksandr Markvart**, **Daria Kulik and Leonid Liokumovich**

Institute of Physics, Nanotechnology and Telecommunications, Peter the Great St. Petersburg Polytechnic University, 195251 St. Petersburg, Russia; markvart_aa@spbstu.ru (A.M.); kulik.dd@edu.spbstu.ru (D.K.); leonid@spbstu.ru (L.L.)
* Correspondence: n.ushakoff@spbstu.ru

**Abstract:** Pulse wave (PW) measurement is a highly prominent technique, used in biomedical diagnostics. Development of novel PW sensors with increased accuracy and reduced susceptibility to motion artifacts will pave the way to more advanced healthcare technologies. This paper reports on a comparison of performance of fiber optic pulse wave sensors, based on Fabry–Perot interferometer, fiber Bragg grating, optical coherence tomography (OCT) and singlemode-multimode-singlemode intermodal interferometer. Their performance was tested in terms of signal to noise ratio, repeatability of demodulated signals and suitability of demodulated signals for extraction of information about direct and reflected waves. It was revealed that the OCT approach of PW monitoring provided the best demodulated signal quality and was most robust against motion artifacts. Advantages and drawbacks of all compared PW measurement approaches in terms of practical questions, such as multiplexing capabilities and abilities to be interrogated by portable hardware are discussed.

**Keywords:** optical fiber sensor; biomedical sensor; optical coherence tomography; intermodal interferometer; singlemode-multimode-singlemode; Fabry–Perot interferometer; fiber Bragg grating; pulse wave; spectral interferometry





## 1. Introduction

Biomedical applications of optical sensing and imaging techniques draw great attention in terms of academic research towards further technological advancements, commercialization of practical devices and search for new applications. Recently, an increasing number of such chronic diseases as diseases of the cardiovascular system, connective tissue dysplasia, diabetes mellitus and others have become very widespread. At the same time, timely diagnosis is the most important stage of treatment and prevention of complications. One of the widely used methods for diagnosing these diseases is pulse wave analysis [1–3], which allows to estimate stiffness of arterial walls and according to it, the state of the cardiovascular system. The advantages of this method include non-invasiveness, high diagnostic confidence, and potential ease of implementation. Typically, high-precision setups for recording pulse wave (PW) signals are stationary and are available only in medical institutions. The examples of such devices include SphygmoCor (Atcor Medical, Sydney, Australia), Complior (Artech Medical, Pantin, France), and others [4,5]. The main factors limiting their implementation in personal and portable devices is their high cost and sensitivity to body movements, which can significantly distort the recorded signals. On the other hand, the development and implementation of portable and personal medical devices based on the registration and processing of pulse wave signals will make it possible to diagnose the above diseases at the earliest stage, significantly facilitating and increasing the effectiveness of their treatment.

A cost-effective solution to PW signal acquisition is offered by photoplethysmography (PPG) [6–8], that detects the change of light intensity transmitted through or scattered from a biological tissue as a response to change of blood volume in blood vessels.

Typically, small vessels such as capillaries are monitored [8,9], however, the original (arterial) pulse wave becomes distorted when coupled from arterial walls to capillary walls, limiting the measurement accuracy. Moreover, PPG is reported to be susceptible to motion artifacts [10,11]. With that in mind, state-of-the-art PPG systems are not ideally suited for high-precision medical inspection, hence, some alternatives might be highly relevant.

Fiber optic sensors are excellent candidates to measure pulse wave signals due to their bio-compatibility, immunity to electromagnetic interference, small footprint of the sensing element, extremely high achievable accuracy and resolution. Several recent papers report the approaches for cardiac monitoring [12–14] and pulse wave measurement [15–20] by means of fiber Bragg gratings (FBG) and Fabry–Perot interferometric (FPI) sensors. A comparison of FBG sensors inscribed in silica and polymer fibers has been reported in [17], with a clear advantage of polymer FBG in terms of achieved resolution. However, humidity sensitivity of polymer optical fiber sensors [21] may lead to undesirable effects in biomedical sensors due to sweating. However, when attached to the skin, such sensors may be subject to uneven deformations, which will distort the shape of their spectral transfer function and reduce the accuracy of the measurements [22,23]. Parasitic sensors deformation may also lead to motion artifacts, deteriorating measurement accuracy.

One way of solving the problem of motion artifacts is increasing the mechanical rigidity of the sensing element. However, for such a weak measurand as pulse wave, this will degrade the sensitivity and might lead to distorted shape of the measured signal. Another solution to the motion artifacts problem might be a sensor, suited for sensing of complex geometries by means of an increased number of variables that can be extracted from the sensor signal. An example of such a sensor is a fiber-optic intermodal interferometer, based on singlemode-multimode-singlemode structure (SMS) [24–27]. In these sensors, a short section of multimode fiber (MMF) is spliced between two singlemode fibers (SMF). Depending on the splicing parameters and type of the multimode fiber, from several to tens of modes are excited in the multimode fiber. After propagation in the MMF section, these modes are coupled to the fundamental mode of the second SMF, leading to intermode interference signal, which encodes the perturbations of the MMF section. Typically, SMS interference signal consists of several interference components, which OPDs, depending on the sensor structure, are related to different perturbations. In addition to the ability of multi-parameter sensing and suppression of cross-talk induced by physical quantities not being measured [24,28], SMS sensors also offer higher strain sensitivities than FBG sensors [26].

Multimode optical fiber sensors are commonly used for multiparameter sensing [29]. An example of simultaneous strain, bending and torsion sensing with a single FBG inscribed in a polymer fiber has been reported in [30]. An example of simultaneous heart rate and breathing rate measurement, demonstrating low susceptibility to motion artefacts is reported in [31,32], wherein the effect of reduced body motion sensitivity was attained by incorporating polymer fiber Bragg grating sensors into a smart textile according to well-defined geometric pattern.

However, mechanical coupling between the pulse wave and even the tiniest fiber-optic sensor unavoidably leads to at least some distortions of the measured signal. Therefore, imaging methods of PW monitoring are considered to be the most advantageous [33]. Optical coherence tomography (OCT) is one of the most prominent approaches of biomedical imaging, enabling to provide a 3-dimensional images of tissues. The basic step of OCT imaging is the so-called A-scan, or axial depth profile of reflection, calculated as the Fourier transform of the measured optical reflection spectrum [34]. 2-dimensional projections and 3-dimensional tomograms are computed by combining the A-scans.

Non-invasive nature of OCT measurements as well as high spatial resolution and ability to perform fast measurements dictated the success of OCT for medical diagnostics. OCT is most widely used in ophthalmology [35–37], as well as endoscopic imaging [38–40], tissue characterization [41,42] and other applications. Initially proposed in the 1980-s [35] originating from white-light interferometry [43], it rapidly matured and became a clinically

accepted tool in 2000-s due to proposal of spectral-domain OCT and in particular swept-source OCT [34,44] and technological advancement of required optical components.

Moreover, phase-sensitive OCT ($\varphi$-OCT) [45,46] is able to measure sub-nanometer displacements of tissues, providing a powerful platform for measuring PW signals with ultimate signal-to-noise ratio (SNR). Up to date, OCT was only reported to be used for PW measurement in retinal arteries [47–49] and relatively small arteries in fingers [50,51], which do not bring much information for cardiovascular system assessment in general due to their peripheral position [33,52].

The aim of the current paper is to compare the performance of fiber-optic sensors with various sensing mechanisms for the task of noninvasive pulse wave monitoring in major arteries (demonstrated on carotid, subclavian and radial arteries). The compared sensing approaches include FPI, FBG, $\varphi$-OCT as well as SMS interferometer. The obtained pulse wave signals were compared on the basis of signal to noise ratio, repeatability and robustness against motion artefacts. It is, to the best of our knowledge, the first comparative study of several types of optical fiber sensors for pulse wave measurement. Advantages and drawbacks of the investigated approaches in terms of signal quality and practical aspects of sensors implementation are discussed. We believe that the reported work will lead to better understanding of applicability of different types of OFS and will pave the way to their widespread use in healthcare.

## 2. Interrogation Setup and Sensors Description

All compared sensing approaches (FPI, FBG, $\varphi$-OCT and SMS) typically utilize spectral interrogation for signal readout. In order to perform an adequate comparison of approaches' performance, the same interrogation setup was used to acquire optical spectra. It consists of Ibsen I-MON USB512 spectrometer (spectrum measurement interval was [1.51; 1.595] μm, variable integration time from 10 μs to 100 μs, spectra acquisition rate up to 3 kHz) and Exalos EXS210066-01 SLED (output power up to 5 mW, central wavelength 1.55 μm, –6 dB spectral width 160 nm, flat-top spectrum shape, the most uniform part coincides with the spectrometer measurement range) installed on Exalos EBD5000 driver board. A photo of interrogation setup is shown in Figure 1a.

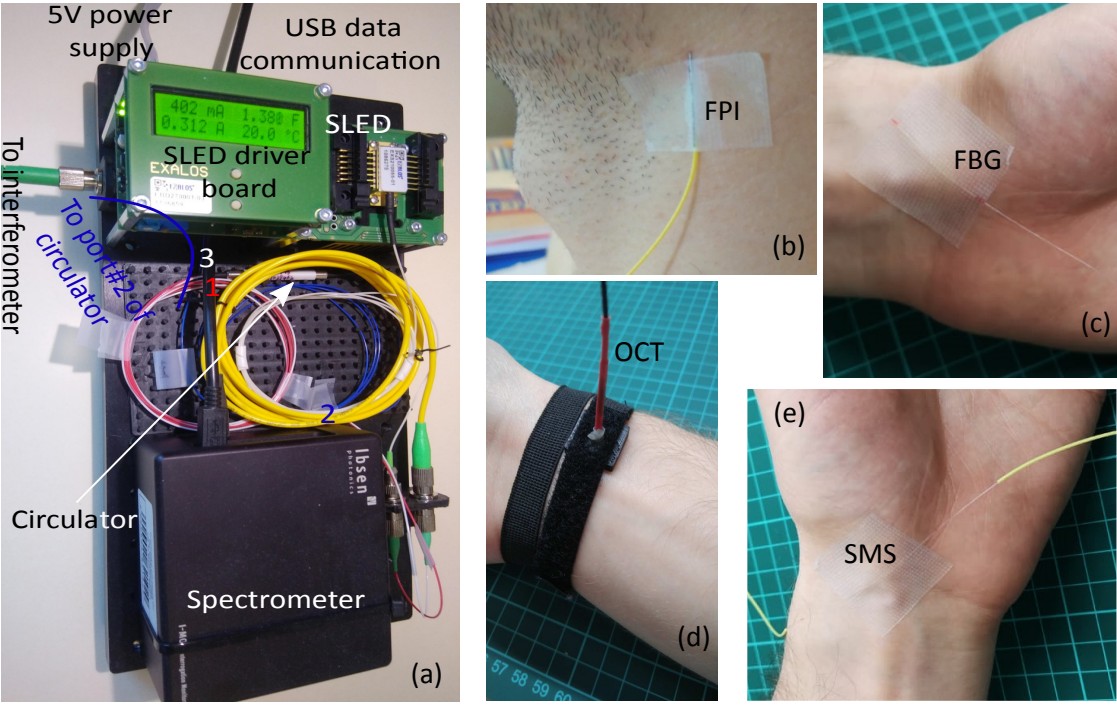

**Figure 1.** Photo of interrogation setup (**a**) and examples of FPI (**b**), FBG (**c**), OCT (**d**) and SMS (**e**) pulse wave sensors fixed over subject's artery. (**a**,**b**) are reproduced from [20].

Depending on the tested sensing element, integration time was varied from 10 μs for FBG and SMS sensors, to 50 μs for FPI and OCT sensors. Output optical power of the SLED was controlled from about 2 mW for FBG and SMS sensors to 5 mW for FPI and OCT sensors by adjusting the operating current. Different parameters of the interrogation system were used in order to match the span of acquired optical spectra to the dynamic range of the spectrometer's analog to digital converter. Details of sensing elements and interrogation system configuration for each of the sensing approaches are listed in subsections below.

### 2.1. Fabry–Perot Interferometric Pulse Wave Sensor

In the case of Fabry–Perot interferometric sensing element, spectra reflected from the sensing element were acquired and further demodulated. The sensing element was connected to the interrogation hardware via optical circulator, as shown schematically in Figure 2a. The sensing element was formed by the end faces of two optical fibers, mated inside a glass capillary. The lead-in fiber was conventional SMF-28 one, while the second fiber was coreless Thorlabs FG125LA fiber, the use of which allowed us to suppress parasitic interference components, originating from light reflection off its far end face. The inner diameter of the capillary was about 130 μm, outer—300 μm, the length was about 1 cm. The fibers were glued at the ends of capillary, the length of the air gap between their end faces was about 57 μm. A metal string about 200 μm in diameter was also glued to the capillary and the SMF fiber 900 μm jacket to increase the mechanical robustness of the sensor.

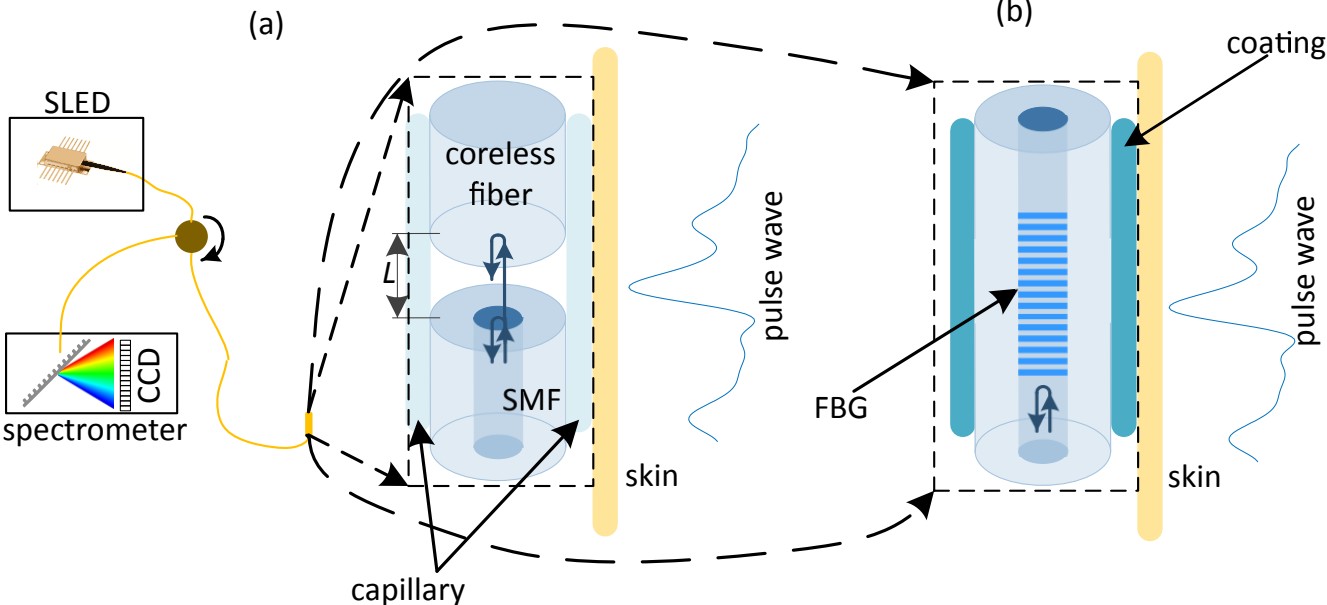

**Figure 2.** Schematic illustration of sensing setups for FPI (**a**) and FBG (**b**) pulse wave sensors.

The sensor was fixed to the skin of the subject, whose pulse wave signal was measured with the use of a flexible adhesive tape over an artery at a point with the most pronounced pulse sensible using fingertips as shown in Figure 1b for carotid artery. No adhesive was added between the skin and the sensor. The sensing principle of such a FPI pulse wave sensor relies upon coupling of the pulse wave to the mechanical perturbation of the sensing element, causing the change of the air gap between the fibers. More details on the FPI pulse wave sensor can be found in our previous paper on this topic [20].

### 2.2. Fiber Bragg Grating Based Pulse Wave Sensor

Fiber Bragg grating is a section of optical fiber with periodic modulation of its refractive index. At each bound of areas with different refractive indexes, the incident optical wave is reflected according to the Fresnel's law. When the period of refractive index mod-

ulation is a multiple of half of the incident light wavelength, all reflected waves add in phase and due to their constructive interference nearly 100% reflectivity can be achieved, while at other wavelengths such interference does not happen and FBG transmits light. Therefore, the reflection spectrum of an FBG has a form of a sharp peak, which position shifts when mechanical perturbation is applied.

The same as for the FPI sensor, the spectra reflected from the FBG were acquired using an optical circulator, as shown schematically in Figure 2c. The FBG was inscribed in a SMF-28 fiber using a KrF excimer laser system with 248 nm wavelength and a phase mask. The central wavelength of the inscribed FBG was 1540 nm and peak reflectivity about 98%. The length of the grating was about 1 cm. Protective UV-cured epoxy acrylate coating with 250 µm diameter was applied to the fiber in order to reduce fragility of the sensor.

The sensor was fixed to skin of the subject, whose pulse wave signal was measured with a flexible adhesive tape over an artery at a point with the most pronounced pulse sensible using fingertips as shown in Figure 1b for radial artery. No adhesive was added between the skin and the sensor. Thanks to smaller weight than the FPI sensor, higher sensitivity could be expected for the FBG sensor, however, it might in fact degrade due to mechanical damping induced by the protective coating.

### 2.3. Phase-Sensitive Optical Coherence Tomography Pulse Wave Sensor

Typically, OCT systems feature a lens, focusing the probing light in a small spot. However, such lenses are quite bulky and can not be robustly fixed over the investigated artery. On the other hand, in order to accurately measure the pulse wave signal, it is advisable to somehow fix the OCT sensing probe above the investigated artery. Therefore, we have chosen to equip the proposed system with a fiber tip probe, similar to those already used in common-path OCT [53,54]. Its advantages are the small footprint of the OCT probe and insensitivity to any perturbations of the lead-in fiber since the interference occurs between the light reflected from the fiber end, which acts as a reference and the light waves reflected at the bounds of different tissue types. This results in optical path differences (OPD) of interference components being unambiguously related to the depth without the need for any calibration and path length alignment.

Due to the small mode field diameter (MFD) of conventional single-mode fiber (10.4 µm at 1.55 µm wavelength), there is a large divergence angle of the output light. In the OCT system this will result in decreased SNR and sensing range. In order to increase the system performance, the single-mode fiber was terminated with a short section (about 2 mm) of graded-index multimode fiber (MMF), spliced to the lead-in SMF fiber. As a result, primarily the fundamental mode of MMF was excited, having larger MFD (16 µm at 1.55 µm wavelength), which helped to decrease the beam divergence inside the tissue. Neglecting the light scattering inside the tissue and adopting a simple model of Gaussian beam propagation, the fraction of light intensity $\eta$ that is coupled back to the fiber can be estimated as [55]

$$\eta(w) = \frac{(\pi n w^2)^2}{(8L\lambda)^2 + (\pi n w^2)^2},\tag{1}$$

where $L$ is the distance the light travelled in the tissue before being reflected, $\lambda$ is light wavelength, $n$ is the mean refractive index of the tissue and $w$ is mode field diameter. The use of MMF for terminating the OCT system also causes light loss due to the mode diameter mismatch, however, in case of relatively large imaging distances the decreased beam divergence due to larger output MFD has greater influence on the increase of reflected intensity. The overall figure of merit (FoM) of using the MMF termination can be evaluated as a ratio of coupling efficiencies $\eta(w_{MMF})$ and $\eta(w_{SMF})$ multiplied by the areas of SMF and MMF modes

$$FoM = \frac{\eta(w_{MMF})}{\eta(w_{SMF})} \cdot \frac{w_{SMF}^2}{w_{MMF}^2}.\tag{2}$$

The FoM values greater than 1 indicate the conditions under which the intensity of light reflected from the tissue is greater in case of MMF termination than in case of simple SMF probe. The dependency of figure of merit on the imaging distance $L$ is shown in Figure 3b in case of wavelength $\lambda = 1.55$ μm, tissue refractive index $n=1.4$ (which is in accordance with the reported data for epidermis refractive index near 1.55 μm [56]), $w_{SMF} = 10.4$ μm, $w_{MMF} = 16$ μm, where it can be seen that the use of MMF termination is effective in case of imaging depth $L > 100$ μm.

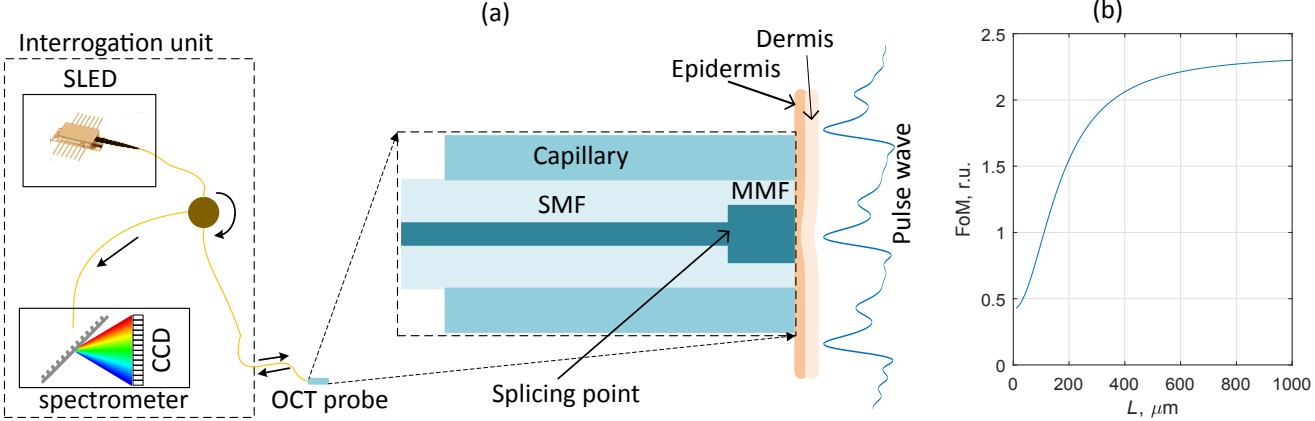

**Figure 3.** Schematic illustration of the proposed OCT pulse wave sensing system (**a**) and dependence of figure of merit on the imaging depth (**b**).

In order to increase the mechanical robustness of the OCT probe, the fibers were packaged in a glass capillary with inner diameter $\sim 140$ μm, outer diameter $\sim 2$ mm and about 5 cm long. The lead-in SMF fiber in the 900 μm jacket was glued to the capillary and protected with a thermally shrinking tube. The fibers were spliced using Jilong KL-280 fiber splicer.

Similarly to the FPI sensor, the spectrum reflected from the OCT probe was measured and further analyzed. The proposed common-path OCT system is schematically depicted in Figure 3a. During pulse wave measurement at the radial artery, the OCT probe was fixed on the wrist using a flexible strap as shown in Figure 1d. During the measurements at carotid and subclavian arteries, the OCT probe was leaned against the skin and held by hand, which did not cause any noticeable signal artifacts.

In contrast to the rest of the pulse wave sensing approaches studied in the current paper, the sensing principle of $\varphi$-OCT sensor did not rely on any mechanical coupling between the pulse wave and the sensing element. In turn, this method was based on the properties of biological tissues, namely on different refractive indexes of epidermis and dermis [56,57], which resulted in clearly observable reflection at their bound. Due to elasticity of biological tissues, the position of this bound with respect to the end face of the OCT probe changed according to the shape of the pulse wave signal. Finally, this change was monitored using phase-sensitive demodulation of the acquired OCT signal.

Because of large attenuation of IR light in spectral range around 1.55 μm by water, contained in biological tissues, the reflection at epidermis-dermis bound mainly contributed to the acquired OCT signals. Additional reflection took place at the bound between stratum corneum and epidermis due to the difference of their refractive indexes. However, since the thickness of stratum corneum was quite small (about 10–15 μm), OPD of the corresponding interference component was small and this component was observed as a slowly oscillating signal component, as in Figure 4b. All lightwaves reflected deeper in dermis could be resolved because of strong attenuation of light with the used spectral range.

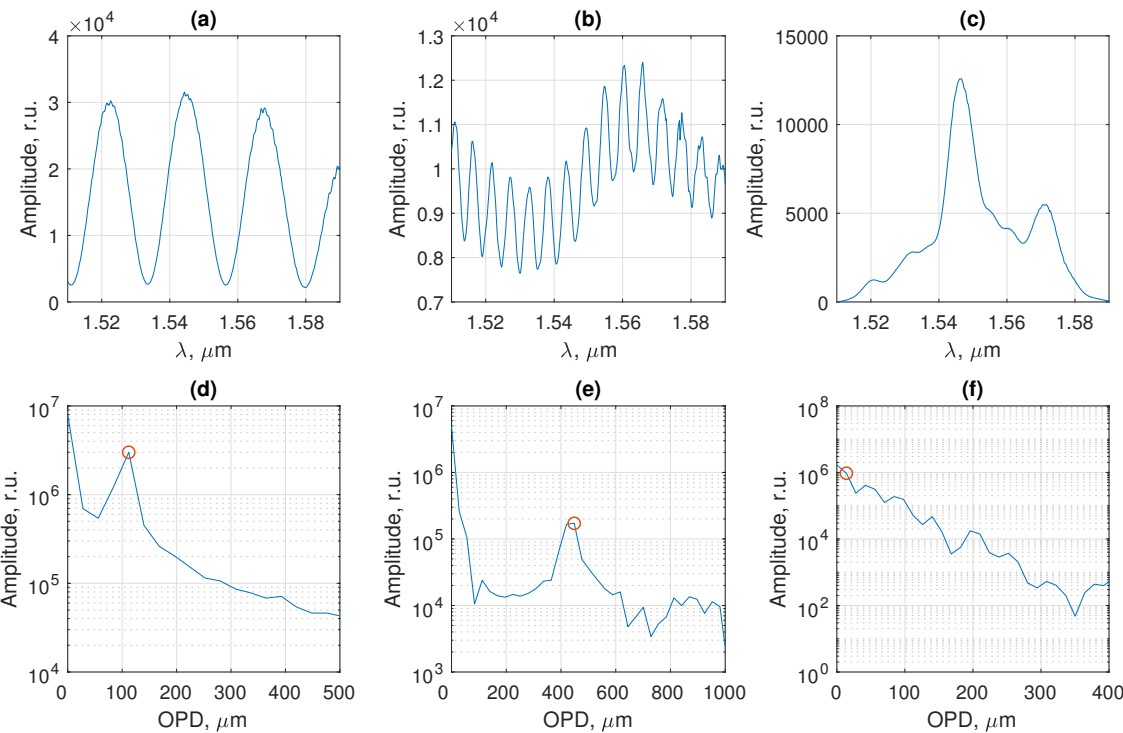

**Figure 4.** Examples of optical spectra and their FFTs of FPI (**a**,**d**); OCT (**b**,**e**) and SMS (**c**,**e**) sensors. Horizontal scales in (**d**–**f**) correspond to optical path length in air (note that OPD of FPI and OCT system is doubled air gap and tissue depth, respectively). Circles in (**d**–**f**) indicate FFT samples, from which phase the target signal was calculated.

Such choice of spectral range might be advantageous in terms of the laser safety as only the tissues close to the surface will be heated, while heat dissipation for the surface tissues is much more efficient than for the deep ones. Moreover, harmfullness of infrared radiation is verified through active use of 1540 and 1550 nm lasers in cosmetology [58]. Due to wide spread of OCT imaging techniques, their safety also has been extensively studied in large cohorts of patients, showing no hazards related to IR radiation exposure [59].

### 2.4. Singlemode-Multimode-Singlemode Interferometric Pulse Wave Sensor

As opposed to sensors described in the previous subsections, when spectral interrogation of SMS sensors is performed, sensors transmission spectra are acquired and processed. Therefore, the scheme of interrogation setup was slightly modified, with the sensor becoming the direct connection between the light source and the spectrometer, as depicted in Figure 5. The SMS sensor was fabricated using Jilong KL-280 fiber splicer. The multimode section consisted of a 5 cm long step-index Thorlabs FG050LGA fiber. At both ends, it was spliced to SMF-28 patchcords. For the sake of mechanical strength of the sensor, protective coating (PC) was left on the MMF and was applied to the splicing points. The reason for using step-index fiber is greater values of phase delays and therefore ability to obtain spectral interference signal with at least several oscillation periods in case of shorter MMF section as compared to graded-index fiber, which is important for practical applications of such sensors.

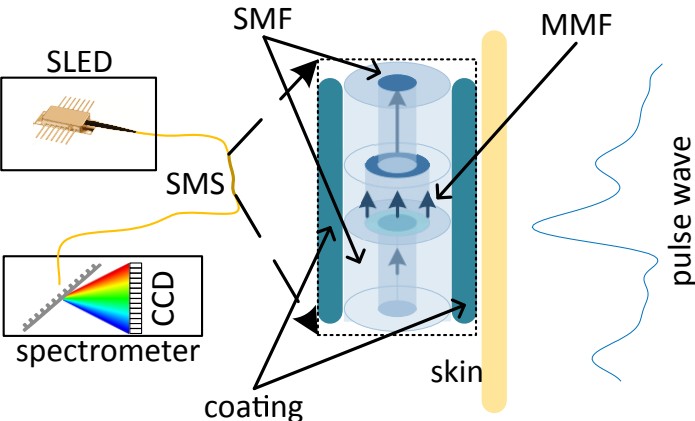

**Figure 5.** Schematic illustration of sensing setups for SMS pulse wave sensor.

The sensor was fixed to skin of the subject, whose pulse wave signal was measured with a flexible adhesive tape over an artery at a point with the most pronounced pulse sensible using fingertips as shown in Figure 1e for radial artery. No adhesive was added between the skin and the sensor. The physical principle of such a sensor relies on mechanical perturbation of the MMF section caused by the pulse wave and consequently induced intermodal phase delays, resulting in a change of acquired spectral interference signal.

## 3. Interferometric and Pulse Wave Signal Processing

The measured optical spectra were stored on the personal computer, to which the spectrometer was connected via USB protocol and further processed in Matlab. FBG signal was demodulated by fitting measured spectra with Gaussian functions as was done in [60,61] with its position, width and amplitude being fit parameters. It was shown in [61] that such the FBG demodulation approach is nearly optimal in terms of demodulated signal noise.

Demodulation of FPI, $\varphi$-OCT and SMS sensors was performed using FFT-based phase calculation, as it is often done in case of spectral interferometry [20,27,62]. Prior to calculating FFT, the measured optical spectrum was interpolated so that spectra samples corresponded to uniform spacing of optical frequency, not the wavelength. In this case interferometer OPD directly corresponded to fringe oscillation frequency, hence resulting in minimal broadening of FFT peaks [63]. Additionally, a Blackman window was applied to the optical spectrum before FFT calculation in order to mitigate the cross-talk between different interference components [64]. In $\varphi$-OCT, the phase (or argument) of a given sample of a complex-valued Fourier transform of the optical spectrum provides the information about the phase of the lightwaves reflected in a given position. Since the axial spatial resolution $\delta L$ of OCT system as limited by the width of the measured optical spectrum, this calculated phase was an average over the waves reflected within a layer of $\delta L$ width, centered around the bound of tissue layers with different refractive indexes. This way, sub-nanometer displacements of various layers in the inspected tissue could be accurately measured, with the resulting displacement resolution determined by the signal-to-noise ratio of the measured optical spectrum.

For FPI and $\varphi$-OCT sensors the phase of the most prominent interference components (marked with circles in Figure 4d,e, which corresponded to FPI cavity and reflections between the OCT probe end face and the bound between epidermis and dermis were used to form the resultant signal. Optical spectra and their FFTs for FPI and OCT sensors (OCT sensor was installed over radial artery) are shown in Figure 4a,b,d,e, respectively. In case of SMS sensor the choice was less obvious since, as can be seen in Figure 4c,f, optical spectrum of the SMS sensor contains a great number of interference components, which was expected for the SMS sensor with a step-index multimode section. Obviously, different harmonic components of the SMS optical spectrum corresponded to interference of MMF modes of different orders. This potentially led to different responses of these components' phases to the PW signal. In this work, we used the phase of the first sample of SMS spectra

FFT (the 0-th sample corresponds to the constant component) as the main demodulated signal for the SMS sensor (marked with circle in Figure 4f) due to its greatest magnitude and therefore, highest achievable SNR of the demodulated signal.

Temporal evolution of the measured vibration signal in interferometric sensors can be observed as a sequence of the measured phase samples. In case of relatively large dynamic range of the vibration signal (if the phase variation of the reflected lightwave is greater than $2\pi$), phase unwrapping [65] must be applied in order to remove the discontinuities of the measured signal. Since the raw phase signal can contain a quasi-static component, caused by breathing, body movements, temperature changes, etc., a high-pass finite impulse response (FIR) filter with stop frequency 0.4 Hz and pass frequency 0.75 Hz was designed and used for phase signal processing.

After the above-mentioned processing of interference signals, pulse wave signals measured with different sensors were obtained. The comparison of sensors performance was carried out according to the following metrics of the pulse wave signal quality: signal to noise ratio (SNR); proportion of successfully extracted pulse wave signal features (PEPWF), such as wave peaks and wave feet; correlation of repetitive pulse wave intervals (CPWI); repeatability of PW interval decomposition features (RPWDF).

The above-mentioned quality metrics were calculated in the following manner: SNR was introduced as a ratio of standard deviation of high-pass filtered signal and white noise level, evaluated from high-pass filtered signal according to [66].

Pulse wave signal features such as wave foot and forward wave peak extraction was performed by one of the algorithms described in [6] (the one developed by *C. Orphanidou*). The algorithm was slightly modified in order to enable successful processing of signals measured from subjects with arterial pulse amplification. The details on algorithm modification and realization details can be found in [20]. Prior to feature extraction PW signal was filtered with a low-pass filter with cut-off frequency 10 Hz. The PEPWF metric was introduced as a ratio of a number of successfully retrieved wave foot and forward wave peak pairs to the total number of pulse wave intervals.

After the pulse wave intervals were successfully identified, it was possible to evaluate the repeatability of PW sensors operation by comparing the waveforms of repetitively measured PW intervals. In order to do that, the signals were divided into intervals corresponding to single pulse waves. The starting points were chosen as the boundaries of the pulse wave slopes. To eliminate the influence of the heart rate on the measurement results, each PW interval was interpolated. As a result, each interval contained the same number of samples, and the sampling rate was normalized on the pulse period. After interpolation, cross-correlation functions of all combinations of pulse wave intervals, identified in the analyzed signal were calculated. The maximal values of these cross-correlation functions were found, their mean value was used as a CPWI metric.

Relatively complex shape of typical PW signal (examples measured at radial and carotid arteries are shown in Figure 6) is due to superposition of direct pulse wave and several reflected waves. While CPWI metric reflects the repeatability of the whole shape of PW interval, more detailed analysis involving feature extraction can provide more quantitative information about PW propagation and sensor performance. In the current paper, we adopted the feature extraction approach, based on approximation of each PW interval by a superposition of six Gaussian functions of a form

$$f(t_0, w, A) = A \exp(-(t - t_0)^2/(w^2)), \tag{3}$$

where $t_0$, $w$ and $A$ are arrival time, width and amplitude of each wave, respectively. Similar multi-Gaussian models were used to describe PW signals in [7,67] and references therein. As a result, each interval was described by 18 parameters (amplitude, position and width of the Gaussian peaks). Initial approximations and restrictions on the values of the approximation parameters were chosen in such a way that the Gaussian functions corresponded to direct and reflected waves in the pulse wave signal, hence providing meaningful physical interpretation of the analyzed PW signals. Pulse wave interval

examples measured using the FPI sensor at the radial artery and at the carotid artery, as well as their approximations by a six-Gaussian model are shown in Figure 6a,b. According to humans' anatomy, the first reflected wave corresponded to the juncture between thoracic and abdominal aorta, and the second reflected wave to the juncture between the abdominal aorta and common iliac arteries [68,69].

Limiting our analysis to quantities with clear physical interpretation, we considered several RPWDF metrics, introduced using the following Gaussian functions parameters:

- standard deviations of delays of the first two reflected waves with respect to the direct wave (denoted as $\text{RPWDF}_{t_i}$, $i = 1, 2$);
- standard deviations of widths of the direct and the first two reflected waves (denoted as $\text{RPWDF}_{w_i}$, $i$ from 0 to 2);
- ratio of standard deviations of the first two reflected waves' amplitudes and the corresponding direct wave's amplitude (denoted as $\text{RPWDF}_{A_i}$, $i = 1, 2$).

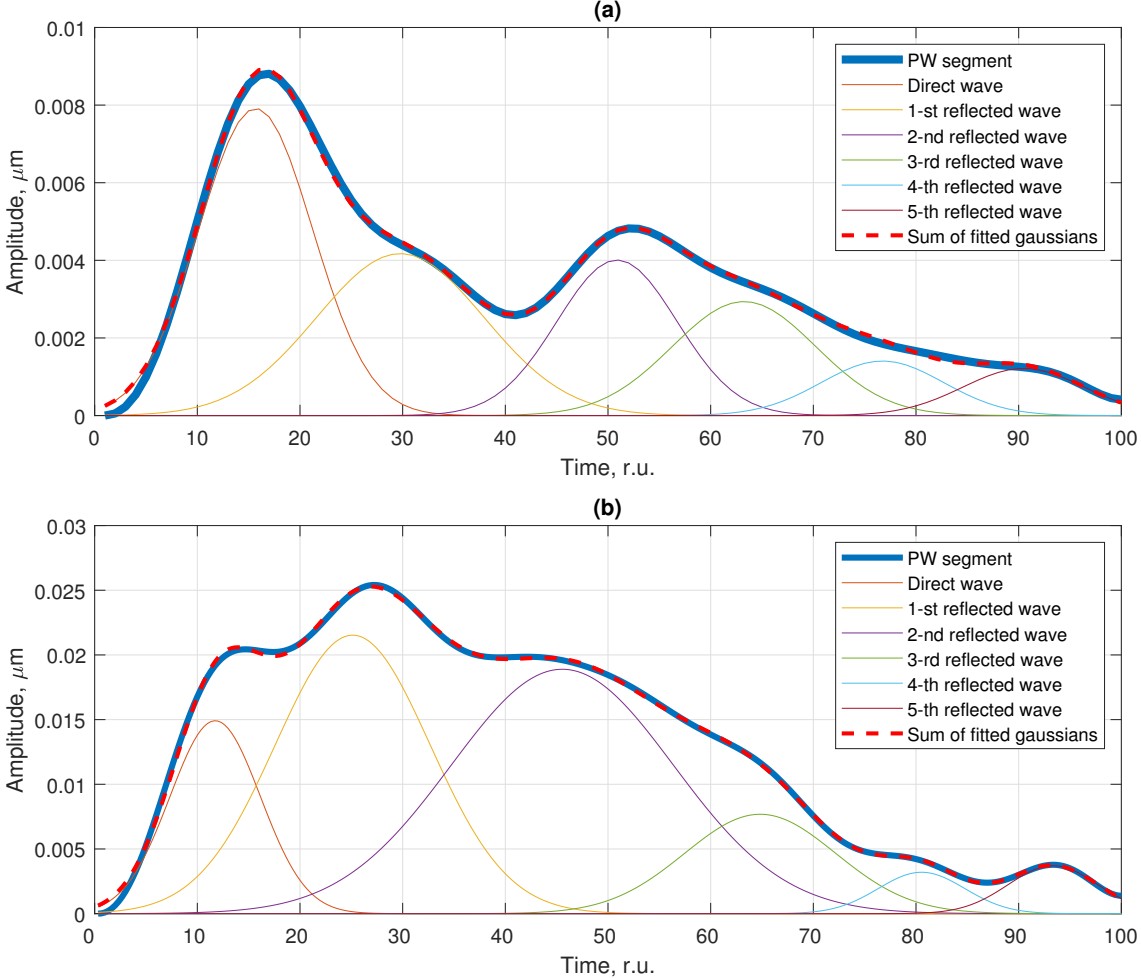

**Figure 6.** Examples of PW signal intervals measured at radial (**a**) and carotid (**b**) arteries and their approximations by a superposition of Gaussian functions, comprising a superposition into direct and reflected waves.

## 4. Experimental Results

The developed system was tested through measuring the pulse wave signals at different arteries, namely, radial, carotid and subclavian. Four individuals participated in the study: a 32-year-old man (Subject I), 26-year-old man (Subject II), 21-year-old woman (Subject III) and a 64-year-old woman (Subject IV). This allowed to demonstrate the system performance via registration of pulse wave signals with different properties. All participants were comprehensively informed about the experimental procedure and gave written

consent before the experiment, which was conducted according to the declaration of Helsinki and approved by the institutional ethics committee.

For each subject, pulse wave signal measurements were performed with each sensing element in 2 min sessions with resting periods between the sessions. The subjects were in a seated position and tried to be as still as possible. Sensing elements were applied as mentioned in Sections 2.1–2.4 at points above the arteries, in which the pulse wave sensed by fingertips was the strongest.

PW signals were demodulated as described in Section 3 using FFT-based phase calculation for interferometric sensors and using Gaussian fitting for the FBG sensor. The above-described metrics were calculated for all PW signals. The variation of the obtained values over different subjects and different arteries did not exceed 5%, so the averaged values were used for comparison of different sensing elements. Such comparison is presented in Table 1, numbers in bold show the best values.

**Table 1.** Comparison of pulse wave signals measured with different sensors. Best scores are shown in bold. Green text color shows the values that deviate not greater than 5% from the best score, orange text color—values which deviation from the best score is between 5% and 25%, red text color—values which deviation from the best score is greater than 25%.

|  | FPI | FBG | OCT | SMS |
|---|---|---|---|---|
| SNR, dB | 65 | 30 | **69** | 40 |
| PEPWF, % | 99 | 95 | **100** | 78 |
| CPWI average, r.u. | 0.91 | **0.96** | **0.96** | 0.81 |
| CPWI $\sigma$, r.u. | 0.07 | 0.05 | **0.03** | 0.12 |
| $RPWDF_{t_1}$, ms | 16 | **7** | **7** | 62 |
| $RPWDF_{t_2}$, ms | 17 | **12** | 14 | 57 |
| $RPWDF_{w_0}$, ms | 4.6 | 9 | **3** | 19.6 |
| $RPWDF_{w_1}$, ms | 10.6 | 8.5 | **6.1** | 16.9 |
| $RPWDF_{w_2}$, ms | 25.6 | **14.7** | 22.5 | 26.4 |
| $RPWDF_{A_1}$, r.u | 0.27 | **0.14** | 0.15 | 0.5 |
| $RPWDF_{A_2}$, r.u | 0.23 | **0.21** | 0.37 | 0.57 |

One of the most important characteristics of any measured signal is signal to noise ratio. In terms of this metric, the best performance was demonstrated by optical coherence tomography (69 dB) and Fabry–Perot interferometer (65 dB). Another objective metric, characterizing signal quality is the amount of signal features, extracted from the measured PW signal using some already validated algorithm, denoted as PEPWF. In this case, three approaches (FPI, FBG and OCT) demonstrated comparable performance, with slight advantage of OCT and FPI over the FBG sensor (100% and 99% versus 95%).

The rest of the introduced metrics demonstrated the reliability of the compared approaches by demonstrating the repeatability of the measured PW signals. Assuming that, for a subject being in a relaxed and calm state and not performing any physical exercises, the shape of the pulse wave signal remains constant, it is possible to compare the shapes of repetitively measured PW intervals and draw conclusions about sensors' performance on this basis. In order to do that, the above-mentioned interpolation of the PW intervals was performed, which eliminated the influence of heart rate variability [70] on signal shape repeatability. For correct results, the length of the analyzed PW signal on the one hand had to be much longer than the pulse period, so that a sufficient number of PW intervals was compared, and on the other hand had to be not long enough so that cardiovascular system state (influenced by fasting, mental stress, etc.) remained the same. Again, the OCT approach demonstrated the best results, with the average cross-correlation of PW

intervals (CPWI average metric) of 0.96 and standard deviation (CPWI $\sigma$ metric) of 0.03. FBG sensor demonstrated the same average cross-correlation, but slightly greater standard deviation of 0.05. A similar picture with the best scores achieved by OCT and FBG sensors could be observed in terms of PW signal features obtained via pulse wave decomposition into direct and reflected waves (RPWDF parameters).

For every sensor, it is important to ensure that the cross-sensitivity to physical quantities that are not the subject of interest. In our case, the main value of interest is sensor deformation (or change of epidermis optical path difference for the OCT approach), caused by the pulse wave. On the other hand, such processes as body temperature change, breathing and body movement can lead to undesired artefacts of the demodulated signal.

For the OCT pulse wave sensing approach, temperature fluctuations resulted in practically no cross-talk since it could only be induced by the epidermis temperature change. However, due to the thermoregulation function of the human body [71,72], skin temperature change occurs slowly and within a finite temperature range. Therefore, any temperature-induced epidermis OPD change is filtered out before the feature extraction procedure and does not result in any influence on the measured pulse wave signal. A similar conclusion can safely be drawn for FBG and FPI sensors—due to their close contact with the skin, high thermal conductivity of silica glass and the compactness of optical fibers, ambient temperature induced only a slight quasi-static component of sensors' signals, which was completely removed by the high-pass filter, used for signal processing. Anyway, the temperature stability of the FPI sensor might be made even better with the use of specially designed temperature-compensating sensing element constructions [73,74]. However, the situation might be slightly different for the SMS sensor, in which the change of temperature can cause intermode phase shifts different from the ones induced by the PW signal, resulting in sensor response nonlinearity.

The effect of sensors cooling on the demodulated PW signal was tested in a separate experiment, in which Subject II participated. After the sensor was applied to the wrist above the radial artery, the sensor was cooled by putting the arm out of the laboratory window. The outside temperature at that moment was about 10 degrees Celsius, leading to sensor cooling. All tested sensors except for the SMS showed no change of demodulated PW signal shape: after filtering out the quasi-static component, the signal shape and signal metrics remained the same. However, cooling of the SMS sensor led to significant change of demodulated signal shape, which is shown in Figure 7. This experiment also validated the repeatability of sensors performance after its reapplication.

Breathing during the PW measurement causes amplitude and frequency modulation of the demodulated signal, as well as adds some low-frequency component due to physiological factors [75]. The low-frequency component is the most detrimental of these effects in terms of further signal processing, however, it is completely removed with the use of high-pass filter, applied to the signal before the feature extraction. The remaining amplitude and frequency modulation of the measured PW signals can be observed in Figures 7 and in 8a,c,e,g.

An additional experiment was performed in order to estimate the susceptibility of different sensors to motion artifacts. Only Subject I participated in this additional study. Sensors were attached to skin over the radial artery near the wrist, as described above. The subject was repetitively squeezing his palm into fist and releasing it with periodicity about 10 s. For each sensor, the PW measurements were performed for 2 min with short rests between the measurements. As can be seen in Figure 8, there were notable motion artifacts and change of the shape of demodulated signal. However, this effect was much stronger for FBG, FPI and SMS sensors than for the OCT sensor. A comparison of the obtained results can be found in Table 2.

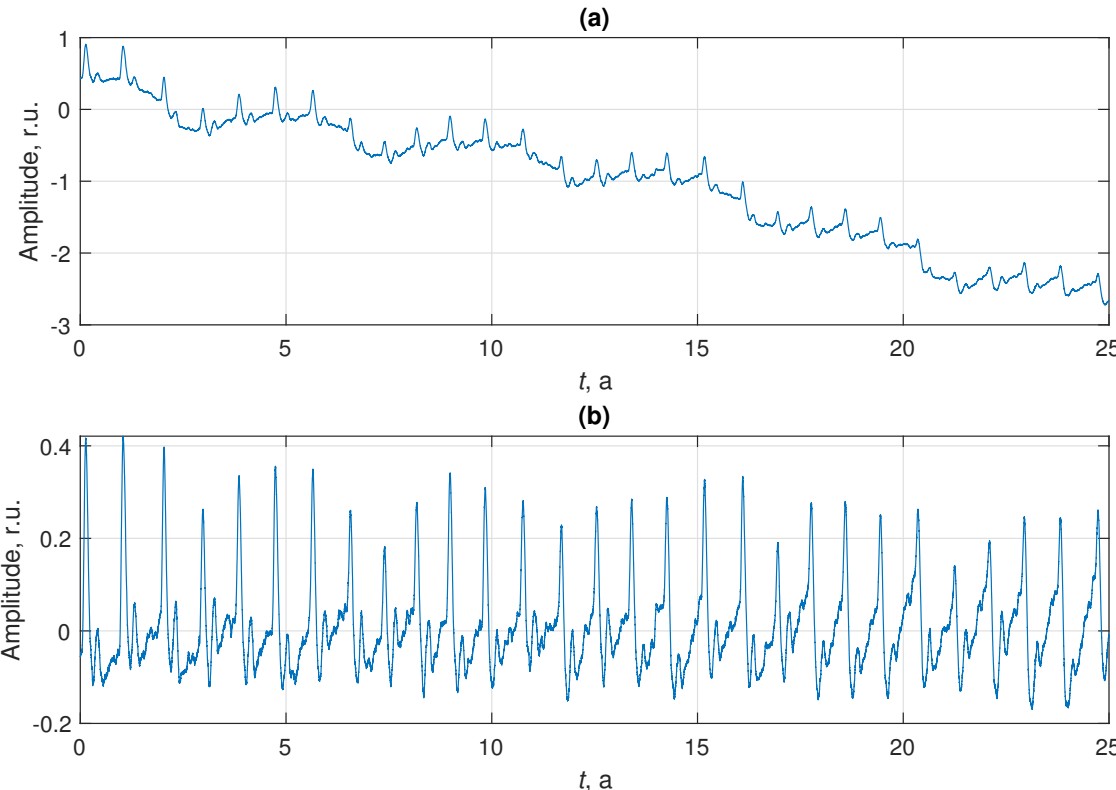

**Figure 7.** Fragment of PW signal measured at radial artery of Subject II with SMS sensor during sensor cooling. (**a**) raw signal, (**b**) high-pass filtered signal.

**Table 2.** Comparison of pulse wave signals measured with different sensors. Measurements were performed at the radial artery with palm movement. Best scores are shown in bold. Green text color shows the values that deviate not greater than 5% from the best score, orange text color—values which deviation from the best score is between 5% and 25%, red text color—values which deviation from the best score is greater than 25%.

|  | FPI | FBG | OCT | SMS |
|---|---|---|---|---|
| SNR, dB | 65 | 30 | **69** | 40 |
| PEPWF, % | 79 | 30 | **88** | 7 |
| CPWI average, r.u. | 0.72 | 0.59 | **0.76** | 0.6 |
| CPWI $\sigma$, r.u. | **0.14** | 0.17 | 0.15 | 0.20 |
| $RPWDF_{t_1}$, ms | 15 | 20 | **11** | 16 |
| $RPWDF_{t_2}$, ms | 22 | 27 | **18** | 23 |
| $RPWDF_{w_0}$, ms | 9.6 | 11.5 | **5.8** | 9.2 |
| $RPWDF_{w_1}$, ms | 10.7 | 7.8 | **6.3** | 9.9 |
| $RPWDF_{w_2}$, ms | 11.4 | 9.4 | 12.4 | **7.6** |
| $RPWDF_{A_1}$, r.u | 0.39 | 0.42 | **0.28** | 0.35 |
| $RPWDF_{A_2}$, r.u | 0.53 | 0.38 | 0.46 | **0.34** |

However, unlike FBG and FPI sensors, the signal of the SMS sensor became considerably distorted only on a relatively short time interval, corresponding to the most rapid palm movement (the corresponding motion artefact can be seen in Figure 8h at time interval

around 6–7 s). However, outside of the motion artifact area, the signal correlated with typical PW shape very well, except for the inverted shape before the 4-th s. Additionally, some fluctuations of PW signal measured with the SMS sensor were present even without any deliberate movements. This indicated nonlinear response of the SMS sensor with simple FFT-based interrogation to the pulse wave.

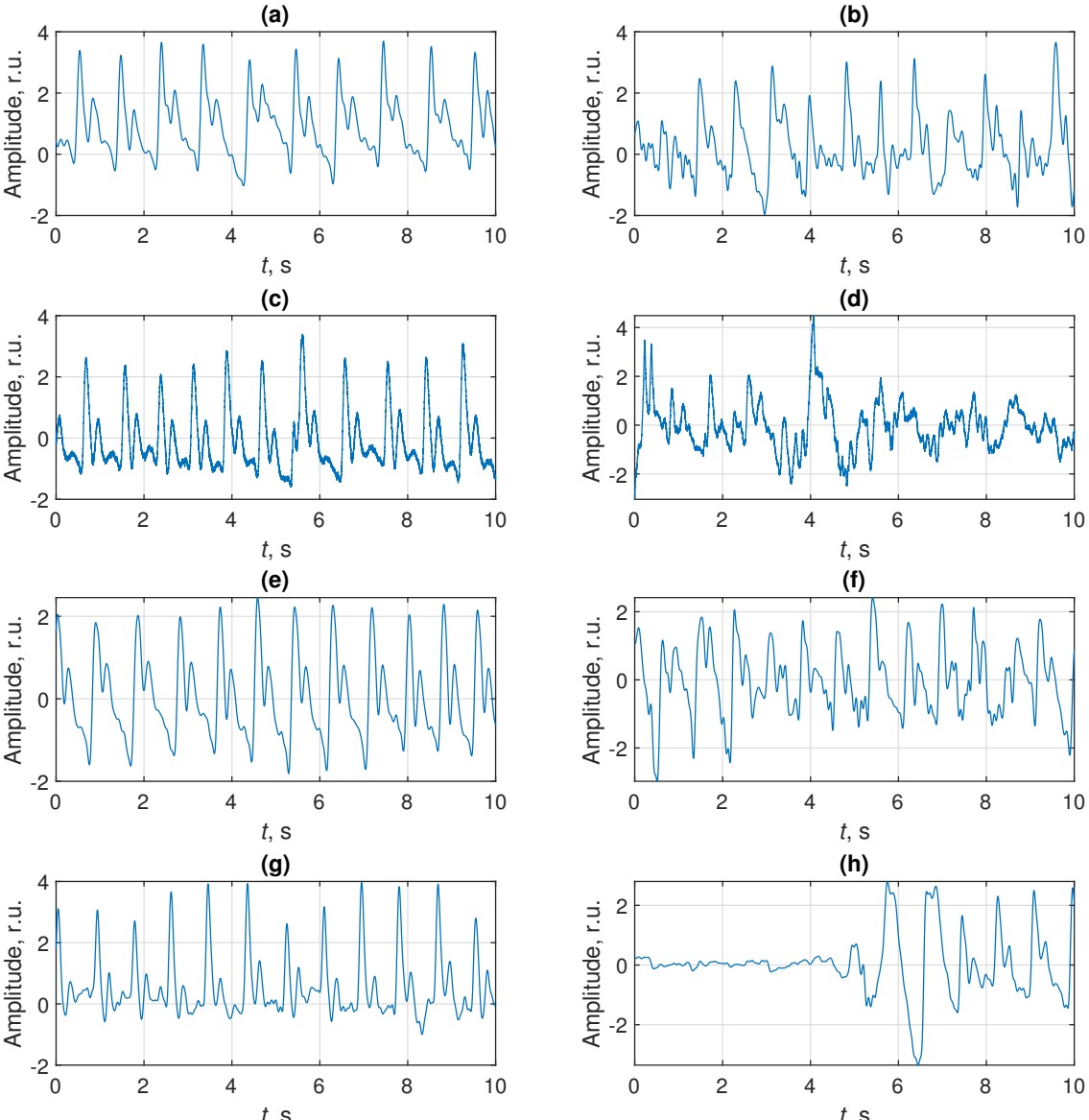

**Figure 8.** Examples of PW signals measured at radial artery of Subject I using FPI (**a**,**b**) FBG (**c**,**d**), OCT (**e**,**f**) and SMS (**g**,**h**) sensors in case of still posture (**a**,**c**,**e**,**g**) and palm movement (**b**,**d**,**f**,**h**).

As can be seen in Tables 1 and 2, there was no obvious winner when the subject was not moving, with OCT method having most of the best metrics and FBG and FPI being close in some of the metrics. However, the PW measurements on radial artery during hand movement revealed the clear advantage of the OCT approach compared to the rest compared ones.

Reproducibility of sensors characteristics was verified through an additional experiment, for which separate sensing elements with similar properties to those mentioned in Sections 2.1–2.4 were used. Subject II participated in this additional experiment, no deliberate body movements were made. The calculated signal metrics were very close to

those reported in Table 1, with the difference not exceeding 5% for SNR, PEPWF and CPWI metrics and reaching up to about 10% for some of the RPWDF metrics.

## 5. Discussion

The results reported in Section 4 demonstrate that the small footprint of the sensor and absence of mechanical coupling between the measured signal and the sensing element are key ingredients for a minimally distorted PW signal. Although spectral interferometric demodulation provides the lowest noises, FPI sensor performance in terms of repeatability is slightly worse than the FBG one, presumably due to greater footprint and non constant mechanical coupling between the pulse wave and the sensor. Nevertheless, an additional experiment performed when the subject made palm movements, showed that the OCT sensor has superior performance in terms of motion artifacts reduction as compared to the rest tested sensors. The main cause for that is the more direct sensing mechanism of the $\varphi$-OCT approach—in contrast to FPI, FBG and SMS, it does not rely on mechanical deformation of the sensing element caused by the pulse wave. Instead, the $\varphi$-OCT approach directly monitors the displacement of biological tissues, which are induced by the pulse wave propagation. This is the main reason for high SNR and measurement repeatability, achieved by the $\varphi$-OCT approach.

Along with the sensors' measurement performance, their ability to be implemented in a portable format, be interrogated by cost-effective hardware and be integrated into modern gadgets is vital for their evolution from laboratory prototypes to commercial products, demanded by healthcare and sport industries. In this regard, there are two important limitations on the interrogation hardware: the use of a spatially incoherent light source, such as LED, which are much more cost-effective than SLED or ASE sources; and low spectral resolution of the optical spectrometer used to acquire sensors' spectra. Another important property is an ability to simultaneously interrogate several multiplexed sensing elements.

Regarding the desirable ability to work with spatially incoherent light, it, in turn, leads to the requirement of only multimode fibers being used in the whole fiber system since it is fundamentally impossible to efficiently couple spatially incoherent light into singlemode optical fiber [76]. This immediately leads to the problem of modal noise [61] that can give rise to parasitic interference signals, which will affect sensor resolution. However, smartphone-based interrogation of FPI [77,78] and FBG [61] sensors has already been successfully demonstrated. All-multimode sensors equivalent to SMS are definitely possible, with intermode interference used for sensing taking place in a largely multimode fiber, sandwiched between two MMFs with smaller number of propagating modes. OCT in all-multimode systems has already been experimentally demonstrated [79], paving the way for its use with portable devices.

Spectral resolution of an optical spectrometer used for sensor interrogation must be in accordance with the width of the sensor's spectrum features, such as reflection peak of FBG or interference fringes of interference sensors. FPI sensors offer the greatest flexibility in this regard, with periods of spectral interference signals ranging from tens of nanometers in case of short-cavity FPIs, allowing to use spectrometers with sufficiently low spectral resolutions of around several nanometers. Typically, the width of the FBG reflection spectrum is about 100–200 pm, therefore, requiring the use of spectrometers with relatively high spectral resolution for correct spectra measurement. However, gratings inscribed in multimode fibers exhibit complex multipeak spectra and therefore are typically chirped in order to obtain a single spectral feature, resulting in a several nm-width reflection peak [61], requiring spectral resolution of spectrometer about 1 nm. This, however, leads to a decrease of achievable measurement resolution and therefore, will result in further decrease of SNR, which, as shown above, even with FBG inscribed in SMF is not quite high. Sensors based on intermodal interference typically have rather complex spectra with fine spectral features, sometimes used for signal demodulation [24]. However, as shown in Sections 3 and 4, the signal of an SMS sensor was obtained from

the interference component, corresponding to the lower-order modes interference. This interference signal component has a large oscillation period and therefore can be acquired using a spectrometer with low spectral resolution (on the order of 10–20 nm). Finally, since the $\varphi$-OCT approach proposed for PW measurement relies on demodulation of phase of a lightwave, reflected from the bound between epidermis and dermis, and is realized using a common-path scheme, it is crucial that the interference signal with OPD equal to doubled optical length of epidermis is acquired without distortions. As follows from the known values of epidermis thickness and refractive index as well as from the obtained data, OPD of such interferometer is about 400 μm. This corresponds to a period of OCT spectrum oscillations about 5 nm, requiring spectral resolution of spectrometer not less than 1–2 nm, which is realizable even with smartphone-based spectrometers [61,80].

In terms of multiplexing capacity, FBG sensors with the use of the wavelength-division multiplexing approach have an obvious advantage. For example, with the spectrometer used in our work with 85 nm spectral span, FBG spectral spacing of 1.5 nm, it is possible to simultaneously interrogate up to 56 multiplexed sensors. FPI sensors also offer multiplexing abilities, typically realized through spatial frequency-division multiplexing (SFDM), sometimes also referred to as OPD-domain multiplexing. Multiplexing tens of FPI sensors is technically possible, but will either require the use of a light source with a relatively high intensity or will result in reduced resolution [81]. Multiplexed OCT systems have already been demonstrated [82,83] with a principle that is similar to SFDM. Limiting factor for multiplexing of OCT probes will be imaging depth and spectral resolution of the spectrometer. In our case of imaging depth up to about 400 μm and spectral resolution of about 200 nm (with corresponding effective coherence length 6 mm and maximal imaging depth of a single probe system of 3 mm) it leads to potentially up to about 6–7 multiplexed probes. To the best of our knowledge, multiplexing of SMS sensors has not been reported yet, although in principle, it can be performed with the SFDM approach as well.

Other practical aspects of such sensing systems include repeatability of sensing characteristics, as well as reproducibility of characteristics of sensing elements and their fabrication cost. Repeatability mainly depends on mechanical coupling between the pulse wave and the sensing element, therefore, the OCT approach has a clear advantage, even despite its small area and seemingly higher requirement to sensor alignment. However, since there is no adhesive used to fix the OCT probe on skin, it is easy to manually adjust it while observing the demodulated signal in order to achieve the highest amplitude. An SMS sensor could also have some advantage in point thanks to its greater length and therefore, greater chance of coincidence of its section with the area of maximal sensitivity. However, this could not be observed in the current study due to the sensor nonlinearity.

In terms of reproducibility, fabrication of FPI sensor of the configuration used in our work is the simplest one in terms of required equipment, requiring a fiber stripper, a fiber cleaver, capillary cleaver (it can also be cleaved manually), glue and equipment for inserting cleaved fibers into capillary (the latter can be performed manually by a trained person). In terms of sensor characteristics reproducibility, the most crucial point is the flatness and cleavage angles of fiber end faces [84,85], which can be monitored right after the cleavage using a conventional optical microscope. For FBG sensor reproducibility of its characteristics solely depends on the grating inscription setup and recoating process and typically is quite high. Reproducibility of the proposed $\varphi$-OCT PW monitoring technique depends primarily on the difference between epidermis and dermis refractive index values, which is consistent within several independent studies [86–88]. The properties of the sensing element mainly influence the amplitude of the demodulated PW signal. Nevertheless, their reproducibility is quite high in case of high quality fiber cleaver and splicer used to produce the sensing element. Reproducibility of SMS sensor characteristics might be the most problematic at the moment due to the above-mentioned sensor nonlinearity. However, its fabrication is not much more complicated than for FPI and OCT sensors, requiring four fiber cleavages and two splices. From the current point, it is hard to predict the cost of sensing elements, however, we believe that for all of them it would be on the same order

of magnitude, since higher cost of grating inscription hardware required to produce FBG sensor is compensated by greater amount of required manual production operations (fiber cleaving, splicing) for FPI, OCT and SMS sensors. Anyway, sensing elements cost still might be several times lower than the cost of even the simplest interrogation hardware.

The above-mentioned parameters of the compared sensing configurations are listed in Table 3. To summarize, we believe that the $\varphi$-OCT approach of pulse wave monitoring may be of the greatest interest for the use in applications where the highest measurement accuracy is needed: medical research and cardiovascular clinics. At the same time, FPI sensors might be most well-suited for use in personal and portable devices since, as follows from the obtained results and the analysis above, they offer a compromise between signal quality and repeatability on the one hand and can be multiplexed, interrogated by simple and cost-effective hardware and demonstrate very weak temperature cross-sensitivity on the other hand. According to the reported results, the main drawbacks of the FBG sensor are relatively low SNR and susceptibility to motion artefacts. However, both of these problems can be solved by the use of FBGs inscribed in polymer fibers, which are also compatible with low-cost interrogation hardware, making them a tight concurrent of FPI sensors, which, according to our study, turn out to be preferable. The use of SMS sensors for pulse wave monitoring and other biomedical tasks may also be prominent if more advanced signal processing algorithms based on detailed analysis of mode propagation in the MMF section are developed.

**Table 3.** Comparison of technical properties of the investigated sensors. $\sigma_{SPEC}$—spectral resolution, $N_{MULT}$—number of multiplexed sensors.

| | FPI | FBG | OCT | SMS |
|---|---|---|---|---|
| Required $\sigma_{SPEC}$, nm | <10–20 | 0.1 | 1–2 | <10–20 |
| $N_{MULT}$ | <10–20 | <50–60 | <6–8 | ? |
| Limiting factor of $N_{MULT}$ | $\sigma_{SPEC}$ | Spectral range | $\sigma_{SPEC}$ | $\sigma_{SPEC}$, number of modes |
| MMF compatibility | + | +, but reduces resolution | + | possible |
| Signal quality rank, still | 3 | 2 | 1 | 4 |
| Signal quality rank, movement | 2 | 3 | 1 | 4 |

## 6. Conclusions

The paper reports on a comparative study of four pulse wave fiber-optic sensors, based on Fabry–Perot interferometer, fiber Bragg grating, optical coherence tomography and singlemode-multimode-singlemode intermodal interferometer. In order to compare the sensors performance, several metrics, evaluating the quality of the obtained PW signals were proposed. According to the scores produced by the proposed metrics, the OCT approach turned out to be the most advantageous. Additional experiment, involving PW measurement of a moving subject also showed that the OCT approach is the most robust against the motion artifacts among the compared ones. However, as follows from the analysis presented in discussion, FPI sensors, having slightly lower scores, are better suited for implementation of portable sensing systems. In turn, the OCT approach of pulse wave monitoring can be successfully implemented for more unique tasks, such as scientific research of pulse wave propagation and cardiovascular system function and be used in specialized clinics.

It should be further noted that configurations of FPI [16,19] and FBG [18,89,90] pulse wave sensors, different from the ones used in our study, have been reported, which might demonstrate different sensing characteristics. Performance of the $\varphi$-OCT approach can be further increased by incorporating micro- and nano-optics for higher light confinement and reduction of light loss inside biological tissues [38,91,92].

**Author Contributions:** Conceptualization, A.M. and N.U.; methodology, A.M. and N.U.; software, D.K. and A.M.; validation, A.M.; formal analysis, A.M.; investigation, A.M. and D.K.; resources, L.L.; data curation, D.K., A.M. and N.U.; writing—original draft preparation, A.M. and N.U.; writing—review and editing, A.M. and N.U.; visualization, A.M. and N.U.; supervision, L.L. and N.U.; project administration, L.L. and N.U.; funding acquisition, N.U. All authors have read and agreed to the published version of the manuscript.

**Funding:** This work was supported by the Russian Science Foundation under Grant 19-72-00051.

**Institutional Review Board Statement:** The study was conducted according to the guidelines of the Declaration of Helsinki, and approved by the Institutional Review Board of Higher School of Applied Physics and Space Technologies (protocol #1 of 12 Februbry 2021).

**Informed Consent Statement:** Informed consent was obtained from all subjects involved in the study. Written informed consent has been obtained from the Subjects to publish this paper.

**Data Availability Statement:** Obtained experimental data are available from the corresponding author upon reasonable request.

**Conflicts of Interest:** The authors declare no conflict of interest.

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
