# Peer review of "Comparison of Pulse Wave Signal Monitoring Techniques with Different Fiber-Optic Interferometric Sensing Elements"

_photonics, doi:10.3390/photonics8050142_

Round 1

Reviewer 1 Report

The authors report on a comparative study between pulse wave signal analyses using different optical fiber sensing platforms. I understand the paper brings interesting information which can be useful for the community working on biomedical sensors. The experiments and results are described adequately. A few minor suggestions are provided in the following to improve the quality of the manuscript.

  1. Please enhance the quality of the diagrams in Fig. 1, 2, 3. Please improve on the placement of the arrows and keep only those that are necessary.
  2. Also, please fix the areas in Fig. 1, 2, and 3, where descriptions and diagrams overlap.
  3. 4 and 5 are referenced in the text before Fig 3. It would be better if the figures appear in the same order as they are referenced in the text.
  4. Probably adding colors in Tables 1 and 2 would help to better identify the best scores.

Author Response

The authors report on a comparative study between pulse wave signal analyses using different optical fiber sensing platforms. I understand the paper brings interesting information which can be useful for the community working on biomedical sensors. The experiments and results are described adequately. A few minor suggestions are provided in the following to improve the quality of the manuscript.

We thank the Reviewer for positive evaluation of our work and for the valuable comments below.

Please enhance the quality of the diagrams in Fig. 1, 2, 3. Please improve on the placement of the arrows and keep only those that are necessary.

Reply: We thank the Reviewer for the helpful comment. We have improved diagrams in figs. 1, 2, 3 (figs. 2, 3, 5 in the revised manuscript) according to their comments.

Also, please fix the areas in Fig. 1, 2, and 3, where descriptions and diagrams overlap.

Reply: We thank the Reviewer for the helpful comment. We have improved diagrams in figs. 1, 2, 3 (figs. 2, 3, 5 in the revised manuscript) according to their comments.

4 and 5 are referenced in the text before Fig 3. It would be better if the figures appear in the same order as they are referenced in the text.

Reply: We thank the Reviewer for the helpful comment. We have moved fig. 4 to the beginning of Section 2 (it is fig. 1 in the revised manuscript) and also moved old fig. 5 (fig. 4 in the revised manuscript) before fig. 3 (fig. 5 in the revised manuscript).

Probably adding colors in Tables 1 and 2 would help to better identify the best scores.

Reply: We thank the Reviewer for the helpful comment. We indicated different scores in tables 1 and 2 with green, dark yellow and red colors.

Reviewer 2 Report

The authors presented comprehensive study on their four optical interferometric sensors for pulse wave signal measurement. The writing structure is clear and inclusive. The simulation and  experimental data are sufficient and interesting for readers. The manuscript could be accepted if the following matters are properly addressed.

1 The impact and novelty should be further clarified. The four sensing devices could be detailed and imaged, beside the schematic illustrations. 

2 Did the authors consider the temperature influence on  the four sensors? This is also crucial for a robust optical sensor.

3  The manuscript is focusing on the authors' four existed sensing devices instead of four general techniques. The title and conclusion of the manuscript could be modified to become more rigorous,as different configurations of four used sensors would lead to different performance.

Author Response

The authors presented comprehensive study on their four optical interferometric sensors for pulse wave signal measurement. The writing structure is clear and inclusive. The simulation and  experimental data are sufficient and interesting for readers. The manuscript could be accepted if the following matters are properly addressed.

Reply: We thank the Reviewer for positive evaluation of our work and for the valuable comments below.

1 The impact and novelty should be further clarified. The four sensing devices could be detailed and imaged, beside the schematic illustrations. 

Reply: We thank the Reviewer for the helpful comment. We have added some comments about potential impact, as well as some comments about the novelty of our work at the last paragraph of introduction and at the conclusion, we have also added about 10 more supporting references. Photos of the sensing elements installed on Subjects are now shown in fig. 1 together with the photo of interrogation setup. We doubt that adding closer photos will add much to the manuscript illustrativeness, since FBG and SMS sensors are basically sections of optical fiber, while internal structure of FPI and OCT sensing elements can't be clearly visualized due to their packaging inside capillaries.

2 Did the authors consider the temperature influence on  the four sensors? This is also crucial for a robust optical sensor.

Reply: We thank the Reviewer for the helpful comment. We have observed slow trends of demodulated signals right after application of sensor to skin, which we attributed to sensor heating. However, in those fragments we didn't observe any change of signal shape. Therefore, we have conducted an additional experiment (please see pages 11-12 and fig. 7 of the revised manuscript for experiment details and some additional comments)

3  The manuscript is focusing on the authors' four existed sensing devices instead of four general techniques. The title and conclusion of the manuscript could be modified to become more rigorous,as different configurations of four used sensors would lead to different performance.

Reply: We thank the Reviewer for the helpful comment. We totally agree with this comment and have added corresponding comment at the conclusion of the revised manuscript. We believe that changing the title in order to reflect this fact will result in its lack of conciseness, and isn't necessary since current title doesn't imply that we have analyzed any sensing structures reported in other works.

Reviewer 3 Report

This paper presents a comparison of Pulse Wave signal monitoring techniques with different optical fiber interferometric sensing elements. An extensive work was done and it is very good to read this paper. I have some comments before final decision.

  1. I miss some important literature about other techniques related with pulse wave and even the fact of heart rate influence, etc. Please read and refer: a) Polymer optical fiber-based sensor for simultaneous measurement of breath and heart rate under dynamic movements, Optics & Laser Technology 109, 429-436, 2019. b) Polymer optical fiber-based sensor system for smart walker instrumentation and health assessment, IEEE sensors journal 19 (2), 567-574, 2019. c) Polymer optical fibers for mechanical wave monitoring. Optics Letters 45 (18), 5057-5060, 2020.
  2. The OCT technique shows better performance and in terms of costs (instrumentation and sensing elements), what is better in terms of cost-quality factor? Please comment and add some words.
  3. The authors claim: " The FBG was inscribed in SMF-28 fiber, its central
    128 wavelength was 1540 nm and peak reflectivity about 98%. The length of the grating was about 1 cm." -  Please add details about how it was inscribed in the fiber, perhaps via UV laser and phase mask, the common technique to inscribe FBG in silica and polymer material optical fibers. Please refer: d)  Narrow bandwidth Bragg gratings imprinted in polymer optical fibers for different spectral windows, Optics communications 307, 57-61, 2013.
  4. The authors claim: "Protective polymer coating with 250 µm diameter was applied in order to reduce fragility of the sensor" - How did you do it? Please add details. Addin protective polymer coating also decrease the sensitivity. Do you use UV resin or other coating?
  5. In terms fo temperature influence, some words need to be added. How we can mitigate it and what is the influence of temperature for each configuration? Please comment and add details in the manuscript.
  6. What is the performance of each proposed sensor in terms of repeatability (making some cycles with the same probe) and  reproducibility (using different probes of each configuration presented in the paper)? Please clarify this point in order to the reader understand in a good way what is really the best option with better performance.

Author Response

This paper presents a comparison of Pulse Wave signal monitoring techniques with different optical fiber interferometric sensing elements. An extensive work was done and it is very good to read this paper. I have some comments before final decision.

We thank the Reviewer for positive evaluation of our work and for the valuable comments below.

1. I miss some important literature about other techniques related with pulse wave and even the fact of heart rate influence, etc. Please read and refer: a) Polymer optical fiber-based sensor for simultaneous measurement of breath and heart rate under dynamic movements, Optics & Laser Technology 109, 429-436, 2019. b) Polymer optical fiber-based sensor system for smart walker instrumentation and health assessment, IEEE sensors journal 19 (2), 567-574, 2019. c) Polymer optical fibers for mechanical wave monitoring. Optics Letters 45 (18), 5057-5060, 2020.

Reply: We thank the Reviewer for the helpful comment. We have modified part of introduction related to motion artefacts suppression and added several references, including the first two of the above-mentioned ones.

2. The OCT technique shows better performance and in terms of costs (instrumentation and sensing elements), what is better in terms of cost-quality factor? Please comment and add some words.

Reply: We thank the Reviewer for the helpful comment. According to our conclusions, currently FPI sensors are preferable comparing to FBG and SMS due to better performance in case of moving Subject and lower requirements to interrogation hardware. Some comments on this matter are added on page 16 at the end of Discussion section, as well as some comments on overall performance of FBG sensors. We have also added table 3, summarizing technical details of the four investigated sensing configurations.

3. The authors claim: " The FBG was inscribed in SMF-28 fiber, its central
128 wavelength was 1540 nm and peak reflectivity about 98%. The length of the grating was about 1 cm." -  Please add details about how it was inscribed in the fiber, perhaps via UV laser and phase mask, the common technique to inscribe FBG in silica and polymer material optical fibers. Please refer: d)  Narrow bandwidth Bragg gratings imprinted in polymer optical fibers for different spectral windows, Optics communications 307, 57-61, 2013.

Reply: We thank the Reviewer for the helpful comment. In this study, we used commercially acquired FBG sensors. Indeed, they were inscribed using UV laser and a phase mask. We have added some comments on how they were manufactured in Section 2.2.

4. The authors claim: "Protective polymer coating with 250 µm diameter was applied in order to reduce fragility of the sensor" - How did you do it? Please add details. Addin protective polymer coating also decrease the sensitivity. Do you use UV resin or other coating?

Reply: We thank the Reviewer for the helpful comment. The coating was UV-cured epoxy acrylate, which is now mentioned in Section 2.2.

5. In terms fo temperature influence, some words need to be added. How we can mitigate it and what is the influence of temperature for each configuration? Please comment and add details in the manuscript.

Reply: We thank the Reviewer for the helpful comment. We have observed slow trends of demodulated signals right after application of sensor to skin, which we attributed to sensor heating. However, in those fragments we didn't observe any change of signal shape. Therefore, we have conducted an additional experiment (please see pages 11-12 and fig. 7 of the revised manuscript for experiment details and some additional comments), which showed that all sensors except for SMS weren't affected by temperature change.

6. What is the performance of each proposed sensor in terms of repeatability (making some cycles with the same probe) and  reproducibility (using different probes of each configuration presented in the paper)? Please clarify this point in order to the reader understand in a good way what is really the best option with better performance.

Reply: We thank the Reviewer for the helpful comment. During this and some of our previous works, we have fabricated about 10 FPI sensors with various parameters and 4 OCT probes, all of them demonstrated very similar performance. However, according to the Reviewer's suggestion , we have performed an additional experiment on reproducibility, mentioned on page 14 at the end of Section 4. We have also added some comments on both repeatability and reproducibility on pages 15-16, section 5.

Round 2

Reviewer 3 Report

The paper can be published